# Biomechanical Evaluation of Temporomandibular Joint Reconstruction Using Individual TMJ Prosthesis Combined with a Fibular Free Flap in a Pediatric Patient

**DOI:** 10.3390/bioengineering10050541

**Published:** 2023-04-27

**Authors:** Krzysztof Dowgierd, Edyta Kawlewska, Kamil Joszko, Jacek Kropiwnicki, Wojciech Wolanski

**Affiliations:** 1Department of Clinical Pediatrics, Head and Neck Surgery Clinic for Children and Young Adults, University of Warmia and Mazury, Żołnierska 18a Street, 10-561 Olsztyn, Poland; 2Department of Biomechatronics, Silesian University of Technology, 44-100 Gliwice, Poland; edyta.kawlewska@polsl.pl (E.K.); kamil.joszko@polsl.pl (K.J.); wojciech.wolanski@polsl.pl (W.W.); 3Scientific Department, ChM Sp Zoo Implants & Instruments, 16-061 Lewickie, Poland; jacek.kropiwnicki@chm.eu

**Keywords:** preoperative planning, custom-design TMJ implant, experimental strength test, FEM analysis, fibular free flap

## Abstract

The main aim of this study was to perform a complex biomechanical analysis for a custom-designed temporomandibular joint (TMJ) prosthesis in combination with a fibular free flap in a pediatric case. Numerical simulations in seven variants of loads were carried out on 3D models obtained based on CT images of a 15-year-old patient in whom it was necessary to reconstruct the temporal-mandibular joints with the use of a fibula autograft. The implant model was designed based on the patient’s geometry. Experimental tests on a manufactured personalized implant were carried out on the MTS Insight testing machine. Two methods of fixing the implant to the bone were analyzed—using three or five bone screws. The greatest stress was located on the top of the head of the prosthesis. The stress on the prosthesis with the five-screw configuration was lower than in the prosthesis with the three-screw configuration. The peak load analysis shows that the samples with the five-screw configuration have a lower deviation (10.88, 0.97, and 32.80%) than the groups with the three-screw configuration (57.89 and 41.10%). However, in the group with the five-screw configuration, the fixation stiffness was relatively lower (a higher value of peak load by displacement of 171.78 and 86.46 N/mm) than in the group with the three-screw configuration (where the peak load by displacement was 52.93, 60.06, and 78.92 N/mm). Based on the experimental and numerical studies performed, it could be stated that the screw configuration is crucial for biomechanical analysis. The results obtained may be an indication for surgeons, especially during planning personalized reconstruction procedures.

## 1. Introduction

Mandibular and temporomandibular joint (TMJ) reconstructions are performed to restore the correct function of the mandible and facial symmetry after resection [1,2,3]. The most common disorders that affect the temporomandibular joint and the mandible, which may lead to resection surgery in the pediatric population, are tumors, injuries, ankyloses, congenital deformations [4,5], and juvenile idiopathic arthritis [6,7,8,9]. The area of reconstruction, the patient’s age, level of skeletal maturity, and the risk of impairment to the opposite joint are the most important considerations in the process of planning the reconstruction of the mandible and temporomandibular joint [10].

Temporomandibular joint (TMJ) reconstruction can be performed using various surgical techniques and materials, depending on the extent of the damage to the joint and the goals of the surgery. Some commonly used ways to reconstruct TMJ include autogenous tissue grafts—these are grafts taken from the patient’s body and used to replace damaged tissue in the TMJ; alloplastic materials—these are synthetic materials used to replace damaged or missing TMJ components; tissue engineering—this involves the use of biocompatible scaffolds that are seeded with the patient’s own cells and implanted in the TMJ to promote tissue regeneration; and joint replacement surgery—in cases where the damage to the TMJ is severe, joint replacement surgery may be necessary. This involves the complete removal of the damaged joint and replacement with a prosthesis made of metal, ceramic, or other materials.

The choice of TMJ reconstruction technique depends on factors such as the patient’s age, the extent of the damage to the joint, and the goals of the surgery. A thorough evaluation by an experienced oral and maxillofacial surgeon is necessary to determine the most appropriate treatment plan for each individual patient.

While the use of temporomandibular joint prosthesis is the most widely used and considered the treatment of choice in adults for TMJ reconstruction with proven satisfactory long-term results, the optimal protocol for TMJ reconstruction in immature patients remains unclear since children frequently have different pathological backgrounds and desired treatment needs. Data from clinical studies have proven the justified use of microvascularized bone grafting techniques in the reconstruction of the resected mandible in children [1]. Commonly performed is TMJ reconstruction of the condyle ramus with autogenous grafts or distraction osteogenesis [11]. Autogenous reconstruction allows continuous growth in the reconstructed joint. For this reason, most are in favor of autogenous reconstruction in childhood [12].

The aim of reconstruction in children is to restore mandibular form and function, reducing the likelihood of revision surgery and encouraging ongoing growth in the reconstructed joint [12]. Reconstruction surgeries performed entirely with the use of fibular free flaps are not functionally adequate, taking into consideration the high risk of developing ankyloses and other complications [13]. Incorrect functioning of the reconstructed temporomandibular joint may also negatively affect the opposite joint, causing damage to the joint, which often requires surgical intervention [14,15].

On the other hand, temporomandibular prostheses currently in use have clearly established indications for use, and their functionality has been proven with clinical results. This type of reconstruction in the adult population is characterized by higher effectiveness and lower complication rates compared to autologous reconstructions, as presented in the literature [16,17,18]. However, such reconstruction of the TMJ has been contraindicated in growing patients because the implants lack the ability to grow [2]. Both of the techniques described above have their specific limitations. The proposed solution in the pediatric population is a combination of both approaches. Therefore, combining a free flap reconstruction with an individual temporomandibular joint prosthesis appears to be a reasonable and indicated solution.

The fixation strength of the free flap and the prosthesis is of key importance to the clinical safety of the use of a medical device. Scientific reports regarding temporomandibular joint prostheses provide results of biomechanical testing on cadaveric mandibles [19,20] as well as numerical predictions with the use of Finite Element Analysis (FEA). For example, the finite element method demonstrates that only three screws are important for the Biomet TMJ fixation [21]. In another study, FEA helped to conduct an analysis and determine which plate provides better stabilization after the segmental mandibular resection [22], the structural optimization of patient-specific TMJ implant [23], and the effect of stress concentration on materials and surrounding tissues [24]. FEA in recent years has been applied to analyze the biomechanical behavior of customized TMJ implants [25,26,27,28], such as the effect of implant design on stress concentration [29,30] and the evaluation of mandibular growth in children [31].

FEA is commonly used to analyze orthopedic biomechanics, such as the biomechanics of plate stabilization on the cervical part of the spine or the human lumbar spine [32,33,34,35,36,37,38,39,40,41,42,43]. However, the literature does not provide biomechanical FEA analysis and testing results of TMJ prosthesis and fibular free flap for both adult and pediatric patients. It is also unclear, as there is no data in the literature, regarding the stability of this system during and after cyclic loads without one or two screws.

The main aim of this study was to perform a complex biomechanical analysis to evaluate the surgical treatment for a custom-designed temporomandibular joint (TMJ) prosthesis in combination with a fibular free flap in a pediatric case. Based on the presented investigations, the surgeon can select the suitable screw configuration for the patient, thereby minimizing surgical complications.

## 2. Materials and Methods

The Committee for Bioethics operating at the Academy of Physical Education in Katowice has approved the analysis of clinical samples.

The patient was a 15-year-old male who underwent bilateral resection surgery of the temporomandibular joint, mandibular ramus, and mandibular body due to fibrous dysplasia. His treatment included simultaneous primary reconstruction using fibular free flaps in combination with individual TMJ prostheses. The prostheses were designed and manufactured based on computer tomography imaging (CT, GE MEDICAL SYSTEMS, 512 × 512 px, slice thickness 0.625 mm) and fitted anatomically to the surfaces of the fibula fragments used as free flaps (Figure 1).

In the planning phase (Figure 1), a computed tomography (CT) scan of the craniofacial skeleton and a computed tomography of the graft site were created. Images were saved as Digital Imaging and Communications in Medicine (DICOM) files. These two-dimensional (2D) DICOM images were then converted into three-dimensional (3D) models in the Standard Tessellation Language (STL) file format.

These models are in the same XYZ coordinate system from which the differences between the virtually planned implant positions before and after surgery can be calculated.

This analysis enables the comparison of pre- and post-operative STL models and informs us of the unfavorable and favorable differences between them.

CAS Mimics software (Materialise, Leuven, Belgium) was used to calculate the arithmetic mean. The difference between the volumes was within ±5 mm.

### 2.1. The Planning/Modelling of TMJ Reconstruction

Prostheses that were identical in size to those designed in the clinical case were manufactured and used for biomechanical testing, with geometrical adjustments to the fibula fragments tested. The design process was as follows:The development of a fibula model in 3D based on CT data (in DICOM format). The 3D segmentation process was performed using Mimics v16 (Materialise Co, Leuven, Belgium).Adjusting prostheses used in the clinical case (composed of the condylar head and mandibular ramus elements) to match the surface of fibular fragments dedicated o biomechanical testing, using Mimics (Figure 2).

The prosthesis was manufactured from cobalt-chromium-molybdenum (CoCrMo) alloy (in accordance with ISO 5832-12) using a 5-axis milling device, choosing the milling strategy and specific tools that allowed us to achieve the highest possible accuracy of the implant. No changes in the length and width of the prostheses and the shape or size of the condylar head element were made. The prosthesis was fixed to the 3D model of the fibula before the beginning of biomechanical testing on human bones to evaluate if the stability and precision of the individual component are the same as in Mimics software. The condylar element was high-gloss polished, and the samples were laser marked (Figure 3).

### 2.2. The Preoperative Evaluation Phase

Biomechanical analysis and testing were performed using fragments of porcine fibulas to evaluate the preoperative TMJ prosthesis model. According to the evaluation of measurements (the graft size providing the implant stability and stable screw positioning) and biomechanical properties (break strength and screw pulling force) of bone fragments commonly used as free flaps, the fibula resembles properties of the mandible the most. A TMJ prosthesis model was a combination of three parts: The fibula, the TMJ prosthesis, and the crews. The analysis was performed for two methods of attachment: a 5-screw configuration and a 3-screw configuration (Figure 4).

### 2.3. Biomechanical Testing—Experimental Study

Samples comprised of an individual temporomandibular joint prosthesis, identical in shape and size to the ones used in the clinical case, and a fragment of bone, to the surface of which the prosthesis was anatomically matched (Figure 5). Prostheses were fixed to the bones with micro screws (System 2.0 × 6 mm, ChM sp. z o.o., Lewickie, Poland), made of Ti6Al4V alloy (in accordance with ISO 5832-3), after previous drilling with a Φ = 1.5 mm drill. Two fixation configurations were evaluated: Using 5 screws and using 3 screws, to determine and compare bone-implant fixation strength of each variant (Table 1). Samples did not differ in any element of the composition, except for the anatomical matching of the prostheses to the individual bone fragments.

Testing was conducted using two methods of sample fixation in the testing machine. In the first method, samples were attached directly to a handle designed specifically for this type of test (Figure 6a), providing the physiological position of the prosthesis at a 120° angle (according to [44], in a population of 1060 patients, the mean value of the mandibular angle, measured based on panoramic radiographs, was 119.97°). The second method included mounting the end of the bone element of the sample in acrylic resin (Electro-Mix cold mounting resin, Metalogis, Warszawa, Poland) and installing the created cylinder-shaped element (Φ = 30 mm) in the testing machine (Figure 6b). The use of two fixation methods allowed us to define the potential effect of strain induced on the bone element because of the fixation of the sample directly onto the handle. Biomechanical testing was conducted using a universal testing machine MTS Insight 100 kN, with a constant rate of traverse of 1 mm/min, in the load-to-failure mode [20].

To calculate and evaluate the stiffness of the fixation the values of the highest peak loads, the highest loads at offset yield and the largest displacement at offset yield were determined. Then, the results of experimental studies were compared with numerical studies and the deviation was obtained.

### 2.4. Finite Element Analysis—Numerical Simulation

Mimics software was used to connect the bone graft, temporomandibular joint prosthesis, and screws, and to define all sets of finite element models. After three-dimensional computer modeling, the models were imported into the MES ANSYS Workbench software (ANSYS, Inc., Canonsburg, PA, USA).

The loading and boundary conditions (Figure 7) were based on previous experimental force research measurements. The loading condition was the combined force (F) acting on the top of the prosthesis (*X*-axis: 0F, *Y*-axis: 0.5F, *Z*-axis: 0.866F). The boundary conditions were to fix the lower surface of the fibula graft without displacement (i.e., *X*-axis, *Y*-axis, and *Z*-axis displacement were set to zero). In addition, to bring this study closer to the actual state, contact between the parts of the model was used. In this study, the contact between the fibula and the screws was defined as bonded. The contact between the temporomandibular prosthesis and the screws and between the temporomandibular prosthesis and the graft was defined as “no separation”. The definition of “no separation” simulates a temporomandibular joint prosthesis attached to the bone surface with no gap between the two parts but with little slippage (Lee, 2014) [45].

It was assumed that the materials of all parts are homogeneous, isotropic, and linearly elastic. Table 2 shows the values of Young’s modulus and Poisson’s ratio applied to the FEM, taken from the literature (Ramos et al., 2019) [20]. Mesh models were created using tetrahedral elements, with convergence tests to control the mesh size performed for convergence and more accurate data. The mesh size in this study was 0.9 mm with approximately 700,000 elements and 900,000 nodes in each group after the mesh.

## 3. Results

### 3.1. Results of the Experimental Tests

Experimental Testing was conducted for seven samples. For samples 1–3, with a five-screw configuration, the highest peak loads (316 N, 185 N, and 397 N, respectively), the highest loads at the offset yield (312 N, 120 N, and 354 N, respectively) and the largest displacement at the offset yield (5.97 mm, 3.08 mm, and 5.08 mm, respectively) were observed. The failure mechanism in all three samples was a transverse fracture of the bone element (Figure 8).

For sample 4, with a three-screw configuration, the highest peak load was observed to be 115 N, the load at offset yield was 112 N, and the displacement at offset yield was 1.33 mm. For sample 5 (three-screw configuration), the highest peak load was 280 N; however, the load at offset yield (52 N) and the displacement at offset yield (1.63 mm) observed for this sample were much lower compared to the results obtained for the five-screw configuration. The failure mechanism for samples 4 and 5 was a longitudinal fracture along the implant, as shown in Figure 9.

For samples mounted in acrylic resin (samples 6 and 7), the highest values of peak load were observed (228 N and 493 N, respectively), as well as high loads at offset yield (154 N and 375 N, respectively); however, the displacement at offset yield values for these samples was lower than in samples installed in the handle (1.67 mm and 3.99 mm, respectively). For sample 6, failure occurred by bone fracture at the resin level and for sample 7, a resin failure occurred. No bone fractures above the resin level and no displacements of the implants were observed (Figure 10). The results of biomechanical testing for all samples are presented in Table 3.

In each case, the peak force measured during the experimental tests was applied as a loading condition in numerical studies. For these conditions, the biggest deviations were observed with the three-screw configuration, and the lowest for the five-screw configuration for samples mounted in acrylic resin. Therefore, the five-screw configuration was assumed to be a better choice, but to prove this hypothesis, the deformation and stress distribution were analyzed for models with human bone properties.

Mounting the samples in acrylic resin instead of directly installing them in the handle eliminates strains on the bone element caused by the handle, as evidenced by the failure mechanism of both samples mounted in resin—failure occurred at the resin level and no other fractures were observed. This type of support is more adjustable for boundary conditions of numerical simulation, which is proved by the lower deviation between experimental and numerical results.

### 3.2. Results of the Numerical Analysis

The fibula bone model, which includes the cortical bone, and the personalized model of the TMJ prosthesis with screws were developed with computer-aided design (CAD) software. The use of human fibula properties for biomechanical analysis allowed us to obtain results precisely projecting the strength of the bone-implant fixation in the patient’s body.

This study referred to the biomechanical testing of the fixation strength of a TMJ prosthesis with a fibular free flap (bone graft). The models of the prosthesis were established with two different screw configurations (Table 4—sample A is a five-screw configuration and sample B is a three-screw configuration). There were seven different sets of calculations.

The total deformation was used to validate the results of numerical studies. The results of experimental tests were compared with numerical studies and the deviation was obtained with the following formula:(1)$$\delta =\frac{∆x}{x}•100\%=\frac{\left|\;x-x0\;\right|\;}{x}\;\;•100\%$$
where *δ* is the deviation, *x* is the result of the test (displacement), and *x*_0_ is the result of the simulation (deformation).

The value of maximum deformation (Figure 11) and von Mises stress (Equivalent Stress, Figure 12) were used as the observation. The total deformation for a model with human bone properties (fibula) under the highest value of load (493 N) was very small (less than 1 mm) for each screw configuration. However, during the experimental tests, the measured displacement was much greater (approximately 4 mm) for this case. The observed stress distribution of the TMJ model shows the maximum stress was located on part of the screws and at the point on the top of the prosthesis, where the force was applied. These effects of applied boundary conditions can be avoided using the Saint–Venant principle, also known as the principle of elastic equivalence. Figure 13 shows the distribution of von Mises stress on the bone (fibula), with the greatest stress located at the corners of the holes in all cases. A comparison of results is presented in Table 5. This table consists of the maximum value of the deformation and the stress with the average value of stress distribution on the bone for each sample.

### 3.3. Postoperative Evaluation of Patient

The patient underwent TMJ reconstruction surgery five years ago to treat fibrous dysplasia that was causing significant pain and other symptoms. The surgery involved the removal of the bilateral lesion and the reconstruction of the joint using an individual alloplastic prosthesis and a fibular free bone graft. The patient has been under observation for five years since the surgery and currently does not report any symptoms or complaints. The patient’s jaw opening and diathesis (presumably referring to the ability to chew solid food) are both normal, which suggests that the surgery was successful in addressing the patient’s TMJ issues. The reported absence of symptoms and normal jaw function suggest a positive outcome for the patient (Figure 14).

## 4. Discussion

Fibrous dysplasia is a benign bone disorder that can affect the mandible condyle, which is the rounded end of the lower jawbone that articulates with the skull. Fibrous dysplasia can cause the bone to become weakened, deformed, and enlarged, leading to functional and aesthetic problems.

There are several reasons why a surgeon may choose to resect fibrous dysplasia from the mandible condyle, including pain—fibrous dysplasia can cause pain and discomfort, especially if it is in the mandible condyle; the pain can be due to pressure on surrounding tissues, nerve compression, or fractures, and surgical removal of the affected bone can help alleviate pain and improve quality of life; functional impairment—fibrous dysplasia in the mandible condyle can lead to functional impairment, such as difficulty chewing, speaking, and breathing; the enlargement of the bone can also cause obstructive sleep apnea, and surgical removal of the affected bone can restore function and improve the quality of life; aesthetic concerns—fibrous dysplasia can cause facial deformities and asymmetry, which can affect a person’s self-esteem and quality of life; the enlargement of the mandible condyle can cause facial asymmetry, jaw protrusion, and malocclusion, and surgical removal of the affected bone can improve facial symmetry, aesthetics, and function; risk of fracture—fibrous dysplasia can weaken the mandible condyle and increase the risk of fractures, especially in cases of trauma. Surgical removal of the affected bone can reduce the risk of fractures and improve stability. In summary, surgical resection of fibrous dysplasia from the mandible condyle may be necessary to alleviate pain, restore function, improve aesthetics, and reduce the risk of fractures. The decision to perform surgery depends on the severity of the condition, the location of the lesion, the age of the patient, and other factors that need to be evaluated by the surgeon.

Taking into consideration the lack of literature data from similar studies, it is not possible to compare the results gained in this study with other studies regarding biomechanical testing of the fixation strength of an individual TMJ prosthesis with a fibular free flap, especially to the mechanics of the bone-implant fixation.

De Maesschalck et al. [46] were the first to describe the methods for assessing the accuracy of hard tissue assessment in mandibular reconstruction using CAS without the need to determine the osteotomy plane using only the superposition tool. Schepers et al. [47] proposed a method for the postoperative assessment of practically planned implants in mandibular reconstruction using CAS by measuring the center point deviation (mm) and the angular deviation (°) per implant.

The main disadvantage of this method is the number of measurements per implant. This reduces the feasibility and results in a loss of accuracy for the entire reconstruction.

In our article, we proposed a simpler method to assess the accuracy and reproducibility of 3DVSP in craniofacial reconstructive surgery in pediatric patients.

The comparison of accuracy results of postoperative computer reconstruction of bone structures is difficult due to the variety of imaging methods and evaluation methodologies used in different studies [9,33,37].

The methodology, which includes the imaging process, the classification of mandibular defects, and the assessment of the volume of three-dimensional (3D) models, allows for the quantitative assessment of accuracy. This assessment can be made by comparing the effect of postoperative reconstruction with the virtual preoperative plan [37,38,39]. Software for computer-aided surgery (CAS) enables matching the preoperative model with the postoperative model [40,41,42,43]. Deviations between the preoperative and postoperative 3D models are then calculated by overlaying the postoperative 3D model on top of the preoperative, virtually planned 3D model.

Experimental models using animal bones, despite many similarities, do not sufficiently depict the bone graft’s reaction to physiologically occurring loading due to micro- and macrostructural differences and bone size. Animal models that show similarities in terms of one parameter are usually vastly different regarding other characteristics. For example, sheep and porcine bones are the closest to human bones in terms of macrostructural view (measurements of long bones), but the density of the cortical bones is much higher than in human bones (human bone—1.64 g/cm^3^, ovine bone—1.85 g/cm^3^, porcine bone—1.70 g/cm^3^ [38]).

The most common animal bone model is the porcine model due to easy access and structural similarities (lamellar bone structure and size) (Kieser et al., 2014) [48]. However, porcine bone is characterized by a much lower Young’s modulus (human bone—14.4 GPa, porcine bone—2.6 GPa) and ultimate whole bone bending strength (human bone—193 MPa, porcine bone—68 MPa), as well as almost triple the value of torsional stiffness (human bone—0.15 Nm/degree/cm, porcine bone—0.38 Nm/degree/cm).

Comparative values for adult human bones were used because values of parameters for pediatric bones are highly variable due to skeletal immaturity, the degree of development, etc., making it impossible to create a universal model of pediatric bones. The use of values for mature bones enables future studies to be compared to those of other authors. Pediatric bones have a higher Young’s modulus, making them less susceptible to fractures [49]. Regarding clinical implications, the developed model of simulation studies indicating the likelihood of biomechanical bone damage still has a safety margin.

Siekaly et al. conducted research on which bone best corresponds to the properties of the mandible. Their study shows similarities in the results of a three-point break test (torque in the mandible—42.49 Nm, torque in the fibula—44.23 Nm) and the screw pull-out force (960 N in the mandible, 638 N in the fibula). The study also evaluated measurements of the fibula, which proved to be sufficient for providing implant stability and correct screw positioning [50].

Experimental tests and complex analysis were carried out using a custom-designed temporomandibular joint (TMJ) prosthesis, manufactured from a cobalt-chromium-molybdenum alloy.

The CoCrMo alloy is a material well-proven in joint prosthesis production, due to its good wear resistance, significantly higher than in titanium. In a study by Royhman et al. [51], CoCrMo and titanium alloy samples underwent tribo-corrosion evaluation in a test setup imitating physiological conditions in the temporomandibular joint. Before and after testing, sample surfaces were examined using interference microscopy. A comparison of the results indicated an increase in surface roughness of 298 nm for titanium and 69 nm for cobalt-chromium-molybdenum alloy. Those results were confirmed by SEM analysis, which showed that wear patterns and crevices were deeper in titanium samples [16].

To determine the mechanical stability of the implants, compression tests including prostheses and fibular free flaps were performed. Thus far, no standardized guidelines were established for testing the fixation strength of an individual temporomandibular joint prosthesis and fibular free flaps. Based on a literature analysis, an experimental method was developed with testing parameters depicting physiological loads occurring in the mandible and temporomandibular joint as closely as possible [21].

Typically, three-screw configuration plates are solid constructs with only three holes in the plate. Such three-hole plates were not tested since the study was based on the real implant used in a real clinical case. Moreover, the biomechanical mechanism of damage that occurred concerns the bone element next to the screw, and not the implant itself, therefore the presence of additional holes in the tested model does not affect the results.

The FEA study investigated the biomechanical effects of different screw configurations inserted into the temporomandibular joint prosthesis. Peak loading on the prosthesis was higher in the five-screw groups and lower in the three-screw configuration groups. Different peak loads are attributed to the length of the lever arm in the temporomandibular prosthesis and the type of specimen fixation when comparing the specimens installed in the handpiece without the acrylic resin specimens. The support was positioned at the bottom of the fibula free flap, and the length of the lever arm between the support and the contact point of the prosthesis and punch was measured. Therefore, the longer the lever arm, the lower the peak load.

The stress on the prosthesis with a five-screw configuration was lower than on the prosthesis with a three-screw configuration. This is assumed to be due to the contact between the screws and fibula free flap and the increasing contact force in the prosthesis plate with the screws [1,52].

With the same force, the fewer screws, the greater the response of the prosthesis and the leaflet free from the fibula. Strain is related to deformation, and Young’s modulus of the components in all groups in this study was constant. Therefore, the greater the deformation of the entire model, the greater the deformation and stress of the model elements. Clinically, the distribution of high stresses on the graft may correlate with a high incidence of peri-implant/screw fractures.

Based on the above analysis, the screw configuration is crucial for this biomechanical analysis (numerical and experimental). The prosthesis with a three-screw configuration has a greater reaction force (stress) on the fibula free flap but less displacement/deformation of the whole model, which may increase the probability of the risk of fracture and osteolysis. The prosthesis plate with a five-screw configuration has less reaction force on the screws and lower stress, which may improve the condition of the osteosynthesis.

The peak load analysis shows that the samples with the five-screw configuration have a lower deviation than the groups with a three-screw configuration. However, in the group with a five-screw configuration, the fixation stiffness was relatively lower than in the group with a three-screw configuration. This can be explained biomechanically by the differences in the fixation stiffness at different mountings of the samples. The samples mounted in acrylic resin caused lower displacement at offset yield values than the samples attached directly to the handle.

Therefore, there is an in-between score for this factor. The clinical connection of the prosthesis to the larger screw is correct and stable while stabilizing the surgical position for immediate action on the leaflets.

The greatest stresses were created at the top of the prosthesis head. This is due to contact with the template in experimental trials, and, meanwhile, with exact numerical use of force application. Any stresses in all models were less than the yield strength of the prosthesis material, with the yield strength of stainless steel and titanium being 700 MPa and 490 MPa, respectively.

## 5. Limitations

This study has limitations, such as the fact that external force is only applied to the top of the head prosthesis. As part of the work, only such stresses and strains of the system in various configurations were discussed. Due to the properties of the material, data from scientific research in the context of work differ from data from experimental research. In this case, only representative bone materials were selected with simplifying assumptions that they are differentiated, isotropic, and linear. Despite the macrostructural and morphological similarities of the porcine fibula and human mandible, the existing differences may limit the universality of conclusions based on experimental results.

Due to the experimental nature of the present research, it was difficult to evaluate which configuration of bone screws is optimal and will not cause the fracture, so it is necessary to perform more tests or simulations.

However, it can still explain the biomechanical difference. FEA of the biomechanical impact of a screw configuration provides a clearer understanding, as well as the basis for a new prosthesis plate design. The research on the prosthesis plate involves not only the general biomechanical effect but also the individual implantation. In the modern techniques of treatment and to increase the patient’s safety, personalized implants are used so the possibility to evaluate the specific force distribution individually for each patient might be interesting for clinical research in the future.

## Figures and Tables

**Figure 1 bioengineering-10-00541-f001:**
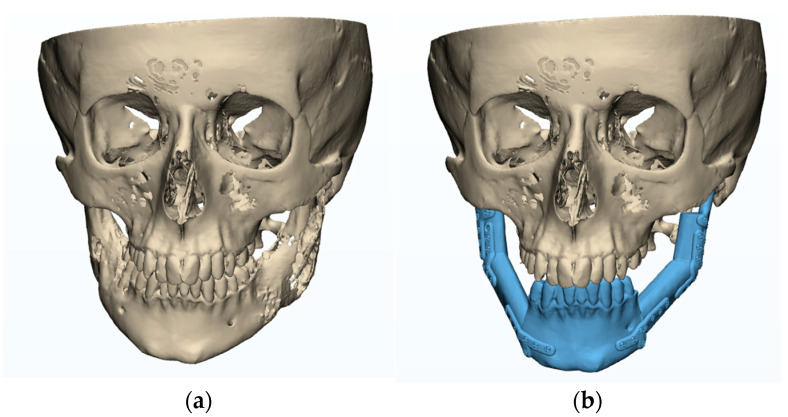
Plan of mandibular reconstruction: (**a**) 3-dimensional preoperative model, (**b**) virtual model of planned customized treatment.

**Figure 2 bioengineering-10-00541-f002:**
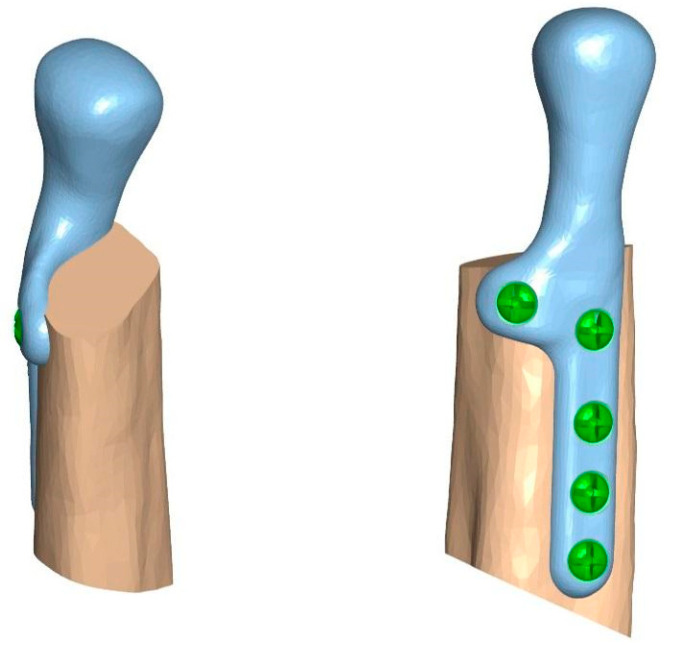
Temporomandibular joint prosthesis matched to a 3D model of the fibula.

**Figure 3 bioengineering-10-00541-f003:**
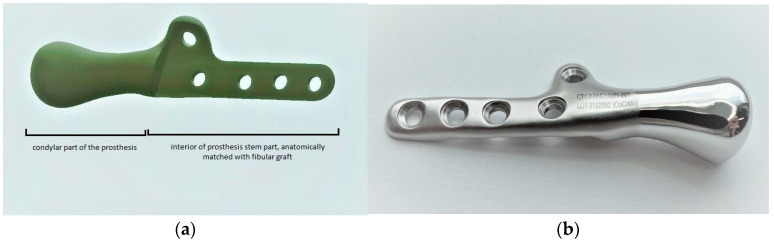
(**a**) 3D model of the prosthesis, (**b**) manufactured TMJ prosthesis (ChM sp. z o.o.).

**Figure 4 bioengineering-10-00541-f004:**
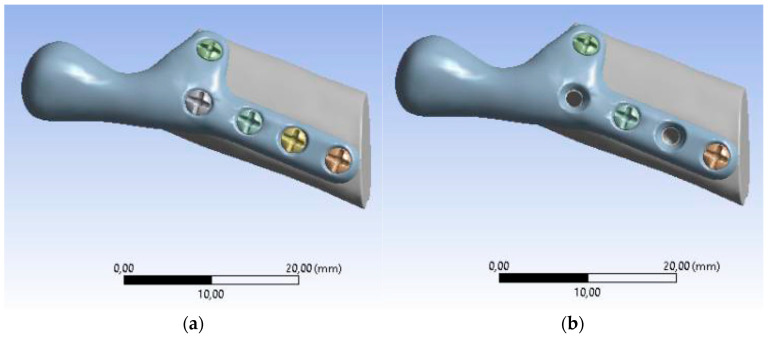
Models of individual temporomandibular joint prostheses: (**a**) 5-screw configuration, (**b**) 3-screw configuration.

**Figure 5 bioengineering-10-00541-f005:**
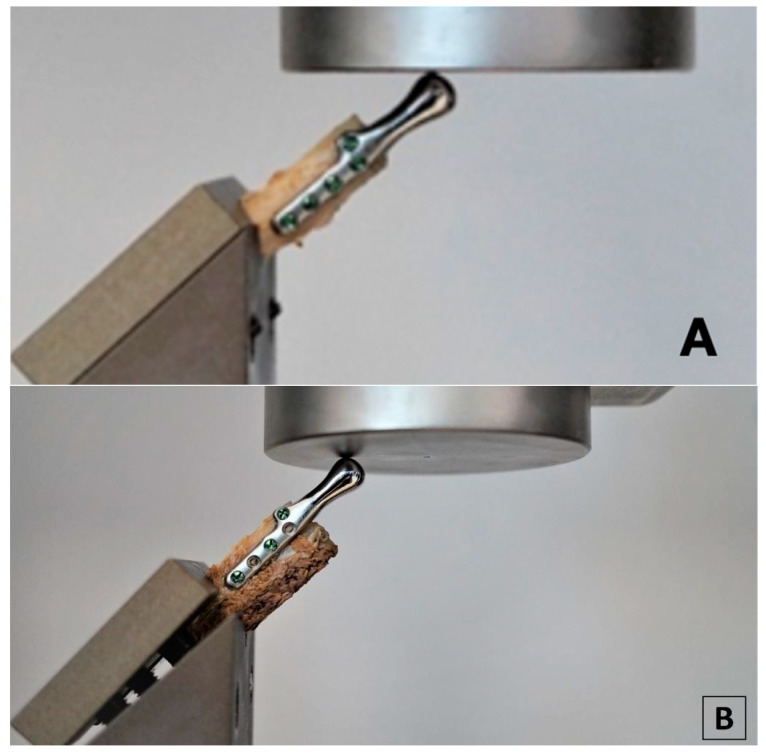
Samples consisting of individual TMJ prosthesis fixed to a fibular free flap; (**A**) 5-screw configuration, (**B**) 3-screw configuration.

**Figure 6 bioengineering-10-00541-f006:**
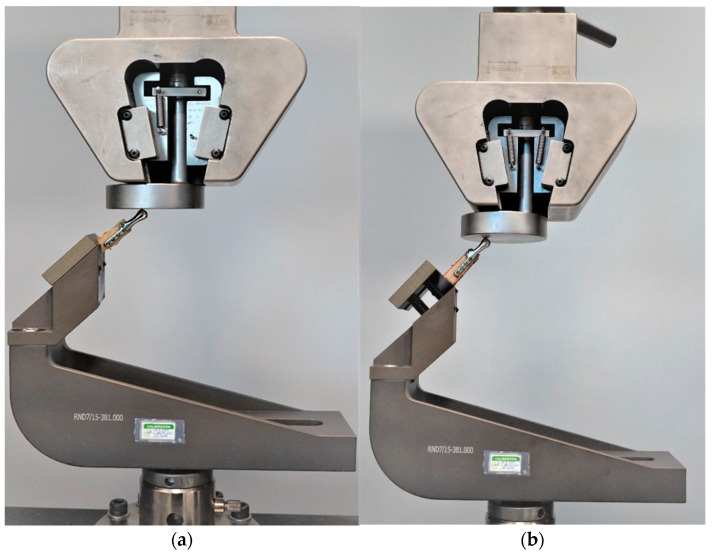
Samples installed in a handle providing anatomical position of the prosthesis; (**a**) sample attached directly in the handle; (**b**) sample mounted in acrylic resin.

**Figure 7 bioengineering-10-00541-f007:**
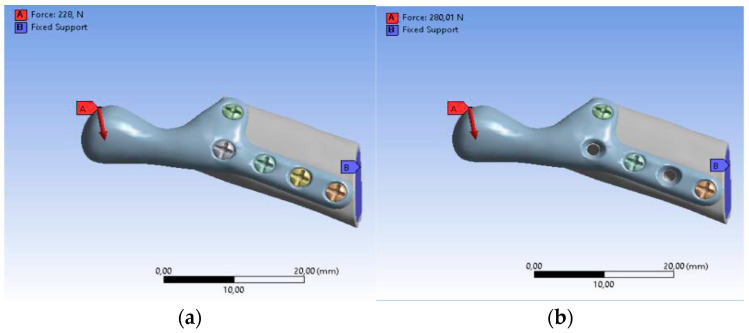
Boundary conditions: (**a**) SAMPLE A, (**b**) SAMPLE B.

**Figure 8 bioengineering-10-00541-f008:**
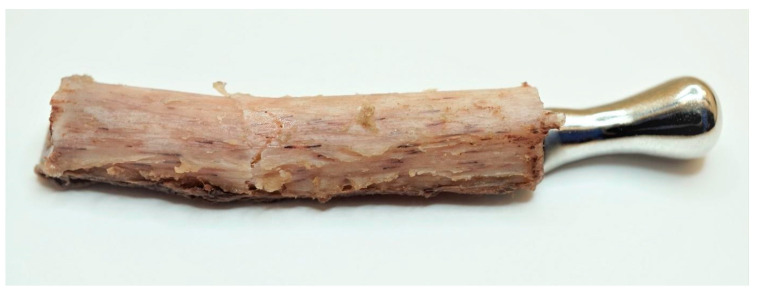
Sample with visible transverse fracture of the bone element.

**Figure 9 bioengineering-10-00541-f009:**
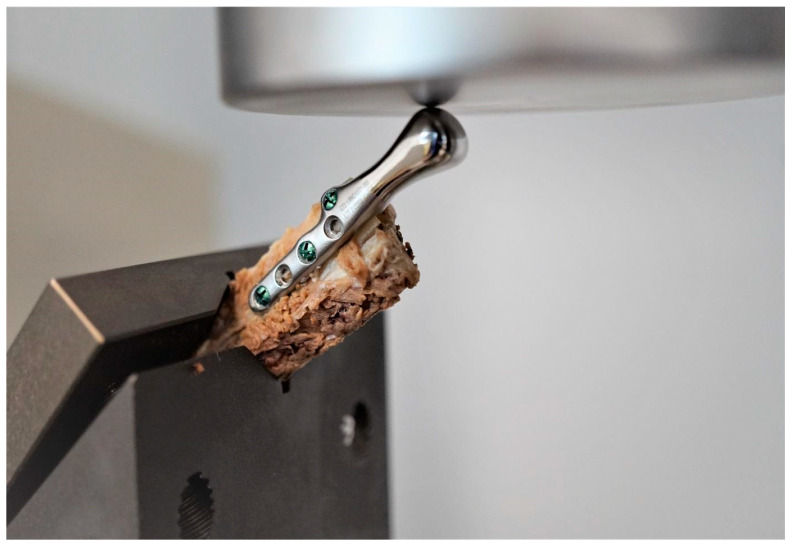
Sample with visible longitudinal fracture of the bone element along the implant.

**Figure 10 bioengineering-10-00541-f010:**
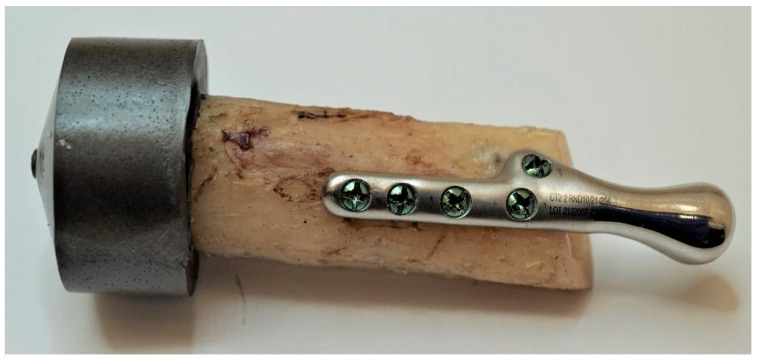
Sample 7, mounted in acrylic resin after testing with no fractures and visible bone displacement from the resin level.

**Figure 11 bioengineering-10-00541-f011:**
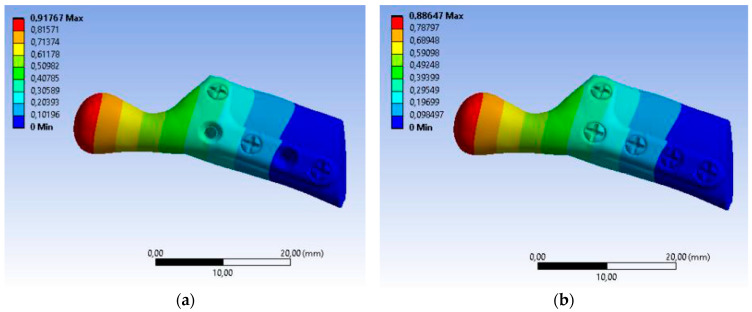
Color map Deformation: (**a**) sample A with 3 screws, (**b**) sample B with 5 screws.

**Figure 12 bioengineering-10-00541-f012:**
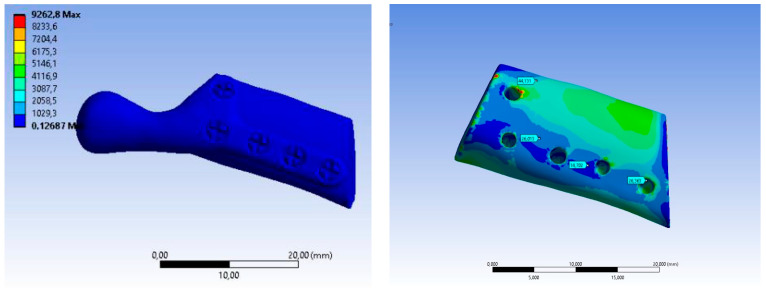
Equivalent Stress SAMPLE A.

**Figure 13 bioengineering-10-00541-f013:**
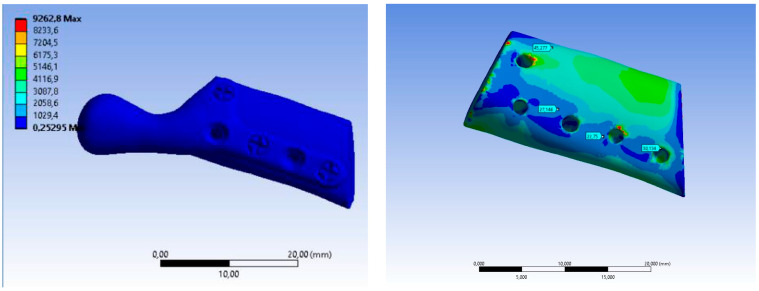
Equivalent Stress SAMPLE B.

**Figure 14 bioengineering-10-00541-f014:**
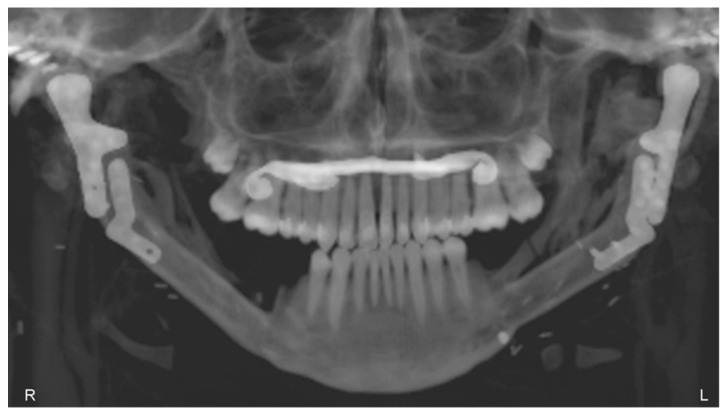
Panoramic X-ray, 5 years after the operation, showing the healed free grafts and both prostheses in the correct position.

**Table 1 bioengineering-10-00541-t001:** Sample sizes.

Sample	Length [mm]	Width [mm]	Cross-Sectional Area [mm^2^]	Number of Screws
SAMPLE 1	48	9.7	53	5
SAMPLE 2	48	9.4	41	5
SAMPLE 3	46.5	14.5	100	5
SAMPLE 4	50.6	10.3	66	3
SAMPLE 5	74.9	14	106	3
SAMPLE 6_B	65.5	16.9	95	5
SAMPLE 7_B	58.3	47.8	183.6	5

**Table 2 bioengineering-10-00541-t002:** The material properties of model components.

Component/ Type of Material	Density	Tensile Strength	Young’s Modulus	Poisson’s Ratio
The fibula/ Human bone	2.1 g/cm^3^	193 MPa	14.4 GPa	0.4
The fibula/graft Porcine bone	1.6 g/cm^3^	68 MPa	2.6 GPa	0.33
TMJ prosthesis/ CoCrMo (ISO 5832-12)	8.4 g/cm^3^	780 to 1280 MPa	210 to 250 GPa	0.3
Screws/ Ti6Al4V (ISO 5832-3)	4.4 g/cm^3^	1000 to 1190 MPa	110 GPa	0.32

**Table 3 bioengineering-10-00541-t003:** Results of biomechanical testing of fixation strength of TMJ prosthesis with a fibular free flap.

Sample	Peak Load [N]	Load of Offset Yield [N]	Displacement at Offset Yield [mm]	Stiffness (Peak Load/Displacement) [N/mm]
SAMPLE 1	316	312	5.97	52.93
SAMPLE 2	185	120	3.08	60.06
SAMPLE 3	397	354	5.03	78.92
SAMPLE 4	115	112	1.33	86.46
SAMPLE 5	280	52	1.63	171.78
SAMPLE 6	228	154	1.67	136.52
SAMPLE 7	493	375	3.99	123.55

**Table 4 bioengineering-10-00541-t004:** Results of TMJ simulation for human bone properties.

Sample (Human)	Equivalent Stress Von-Mises Maximum (Average—Bone) [MPa]	Maximum Deformation [mm]
SAMPLE A	9262 (50.05)	0.91
SAMPLE B	9262 (49.86)	0.88

**Table 5 bioengineering-10-00541-t005:** Comparison of the results of FE analysis performed for seven variants of loads and boundary conditions.

Sample	Experimental Test		Numerical Study		The Deviations $\frac{\left|x-x_{0}\right|}{x}\cdot 100\%$ [%]
	Peak Load [N]	Displacement Yield—*x* [mm]	Maximum Equivalent Stress Implant (Bone)[MPa]	Maximum Deformation—*x*_0_ [mm]	
SAMPLE 1	316	5.97	5984 (49.98)	5.32	10.88
SAMPLE 2	185	3.08	3503 (29.26)	3.11	0.97
SAMPLE 3	397	5.03	7517 (62.79)	6.68	32.80
SAMPLE 4	115	1.33	2355 (18.06)	2.10	57.89
SAMPLE 5	280	1.63	5261 (35.78)	2.30	41.10
SAMPLE 6	228	1.67	4283 (29.31)	1.76	5.38
SAMPLE 7	493	3.99	9262 (63.38)	3.81	4.51

## Data Availability

Not applicable.

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
