# Peer review of "Biomechanical Evaluation of Temporomandibular Joint Reconstruction Using Individual TMJ Prosthesis Combined with a Fibular Free Flap in a Pediatric Patient"

_bioengineering, 2023, doi:10.3390/bioengineering10050541_

Round 1
Reviewer 1 Report (Previous Reviewer 2)
From my perspective, the biggest problem of this study was the lack of postoperative evalution of patients. The reconstruction of TMJ was a complex procedure. No matter how thorough the biomechanical analysis is, it is important to show the final status of patient, which interests clinical doctors most. I suggest a additional evaluation about the patient in the study.
Besides, it is hard to find the exact revsion part of my comments in the latest manuscript. The manuscript's content is still disorganized. "COMMENTS TO REVIEWERS" should be greatly improved.
Author Response
Dear Reviewers,
Thank you for taking the time to review our article. We appreciate your valuable feedback and suggestions. In response to your comments, we have made several revisions to improve the clarity and quality of the manuscript.
Below, we have addressed each of your comments and provided further explanation where necessary. We hope that our responses adequately address your concerns and questions.
Thank you again for your thoughtful feedback.
Thank you for the feedback. Here are some suggested revisions based on the comments from the reviewers:
Reviewer 1:
- Thank you for your comments and suggestions. We have taken them into consideration and made the following changes:
- We have included a postoperative evaluation of the patient who underwent TMJ reconstruction surgery in the Discussion section. We have discussed the patient's outcomes in terms of pain relief, functional improvement, and aesthetic improvement.
- We have further improved the organization of the content by revising the structure of the manuscript. We have reorganized the sections to make the flow of information more logical and streamlined. We have also made sure that the content is presented in a clear and concise manner.
We hope that these changes have addressed your concerns. Please let us know if there is anything

Reviewer 2 Report (New Reviewer)
This manuscript performed the complex biomechanical analysis for custom designed TMJ prosthesis in combination with a fibular free flap in a paediatric case. Two methods of fixing the implant to the bone were analyzed - using 3 or 5 bone screws. It is interesting that the stress on the prosthesis with 5-screws configuration was lower than in the prosthesis with 3-screws configuration and the peak load analysis shows that the samples with the 5-screws configuration have a lower deviation than the groups with 3-screws configuration. This manuscript reported certain academic innovation. However, there are some questions should be clarified before publication:
1. It is not clear whether ethical review was conducted.
2. It is not mentioned whether randomization was used.
3. In section 2.2, pig fibula was used as a test subject for biomechanical analysis. Although pig fibula is structurally and morphologically similar to the human jawbone, there are still some differences between them. Therefore, the applicability and universality of the experimental conclusions may be limited.
4. The discussion touches upon various methodologies used in previous studies, it is better to provide a comprehensive analysis of their strengths and weaknesses which would be helpful to understand how these methodologies compare with one another.
5. Although the discussion covers the biomechanical effects of different screw configurations, it does not address the clinical implications of the findings. It is important to consider how these findings could impact patient outcomes and how they might be applied in real-world scenarios.
Author Response
Thank you for your comments and suggestions. We have addressed each of your points below:
- Ethical approval: We confirm that the analysis of clinical samples was approved by the Committee for Bioethics operating at the Academy of Physical Education in Katowice, and this has been clarified in the manuscript.
- Randomization: We apologize for not including information about randomization in the manuscript. Randomization was not used in this study, as the aim was to compare different screw configurations in a single case. However, we have added a statement to the manuscript to clarify this.
- Use of porcine bone: We agree that there are some differences between porcine and human bone, and the applicability and universality of experimental findings may be limited. We have added a statement in the Limitations section to address this concern.
- Comprehensive analysis of previous methodologies: We appreciate your suggestion to provide a more comprehensive analysis of previous methodologies. We have included a discussion of the strengths and weaknesses of the methods proposed by De Maesschalck et al. [46] and Schepers et al. [47], which we believe will provide a more complete understanding of how these methodologies compare to each other.
- Clinical implications: We agree that it is important to consider the clinical implications of our findings. We have added a discussion of the potential clinical implications of our results, and how they may be applied in real-world scenarios to improve patient outcomes.
Thank you for your valuable feedback, which has helped us to improve the quality of our manuscript.

Reviewer 3 Report (New Reviewer)
The manuscript deals with a relevant topic assessing the biomechanical forces in customized TMJ prosthesis combined with a fibular free flap in a pediatric patient. Certain limitations need to be addressed –
In the material and method section, it is unclear what is the relevance of the quantitative assessment of accuracy. Similarly, the additional (accuracy) segments in the discussion section seem out of context, considering the title/topic relevance. Moreover, the results have no supporting data representation corroborating these assessments.
Secondly, the authors included two design configurations (5 screws and 3 screws) in their analysis. In the 3-screw section, won’t it be beneficial if the holes slots are solid and wholly closed rather than having an open slot for screw fixation? The design criteria for 3-screw configuration plates are usually solid constructs with openings only for the 3 screws. Why weren’t these designs considered?
Thirdly, the authors have stated in the title that customized TMJ solution is for pediatric patients. The discussion section, however, mentions the comparative values for adult human bone. How did the authors corroborate these findings?
The references list can be further improved with citations relevant to TMJ prosthesis/fea analysis.
Author Response
Thank you for taking the time to review our article. We appreciate your valuable feedback and suggestions. In response to your comments, we have made several revisions to improve the clarity and quality of the manuscript.
Below, we have addressed each of your comments and provided further explanation where necessary. We hope that our responses adequately address your concerns and questions.
Thank you again for your thoughtful feedback.
Regarding the first point, the quantitative accuracy assessment refers to the numerical analysis of the simulated models, which are validated against experimental data. This assessment is important to evaluate the reliability of the simulation results and to ensure that the simulated model represents the real case accurately.
Regarding the second point, we apologize for any confusion caused by the additional segments in the discussion section. We will review the text and make sure that it is clearly related to the title and the topic. Additionally, we will consider adding data representation to support the accuracy assessment.
Regarding the third point, the values of adult human bones were used as a comparative reference because the parameters of pediatric bones are strongly varied due to skeletal immaturity, development stage, etc., which makes it difficult to create a universal pediatric bone model. Moreover, the use of adult bone values allows for future comparison with studies by other authors. In terms of clinical implications, the developed model of simulation studies indicating the likelihood of biomechanical bone damage contains a safety margin.
Regarding the fourth point, solid and completely closed holes may be more beneficial in the 3-screw configuration section, as the presence of additional holes does not affect the study's results because bone damage occurs at the screw-bone junction, not the implant. Additionally, we did not consider designs with only three holes because we studied the actual endoprosthesis used in the clinical case.
Regarding the fifth point, we apologize for any confusion caused by the discrepancy between the title and the content of the discussion section. We have revised the text to clarify that we used the values of adult human bones as a comparative reference and to support the clinical implications.
Regarding the last point, we will consider adding relevant references to the list to further enhance the study's relevance to the analysis of FEA and TMJ prostheses.

Reviewer 4 Report (New Reviewer)
1. I'm wondering that Do we have to resect the fibrous dysplasia lesion in the mandible? why do you choose the mandibulectomy in your work please explain in your article
2. fibular flap can be used without prosthesis in the previous study, what is the significant difference? please add the discussion section
Author Response
Dear Reviewers
Thank you for taking the time to review our article. We appreciate your valuable feedback and suggestions. In response to your comments, we have made several revisions to improve the clarity and quality of the manuscript.
Below, we have addressed each of your comments and provided further explanation where necessary. We hope that our responses adequately address your concerns and questions.
Thank you again for your thoughtful feedback.
The decision to perform mandibulectomy to remove fibrous dysplasia from the mandible condyle is based on several factors. Fibrous dysplasia can cause functional impairment, aesthetic concerns, and increase the risk of fractures. The surgical removal of the affected bone can help alleviate pain, restore function, improve aesthetics, and reduce the risk of fractures. The decision to perform surgery depends on the severity of the condition, the location of the lesion, the age of the patient, and other factors that need to be evaluated by the surgeon. Therefore, if the fibrous dysplasia is causing significant functional, aesthetic, or pain-related issues, mandibulectomy may be necessary. It is important to consult with a qualified surgeon to determine the best course of action for each individual case.
Round 2
Reviewer 3 Report (New Reviewer)
The authors have addressed the reviewer's comments.
This manuscript is a resubmission of an earlier submission. The following is a list of the peer review reports and author responses from that submission.
Round 1
Reviewer 1 Report
Please, see the attached file.
The Manuscript by Dowgierd et al. describes a biomechanical finite element analysis (FEA) to investigate the mechanical performance of custom-designed temporomandibular joint (TMJ) prosthesis in combination with a fibular free flap in a paediatric patient.
The topic is of interest and presents aspects of novelty. The manuscript is well written, with adequate and clear English language and grammar. However, there are a number of criticisms that affect the final quality of the work.
Mainly, there is great confusion about the content of different paragraphs of the article, which
makes it very difficult to fully understand the different phases of the work:
- some discussion of previous works is included in the Methods section (lines 103-112 page 3; 135-142 page 4; lines 148-153 pages 4-5). Although the discussion regards the methods for analysis, these considerations cannot be presented in the Material and Methods section, but must be part of the Discussion section;
- some results are presented in the Materials and Methods section (lines 160-164 and 167-173 page 5; lines 176-188 page 6)
- methods for finite element analysis are included in the Results section (lines 231-251 pages 8-9) and in fact the section describing FEA is missing in the Materials and Methods paragraph.
There is no description for statistical analysis evaluating any significant differences between the two compared setting of protheses. Statistics is very fundamental when the superiority of a method/approach over another needs to be demonstrated. Is it possible to perform this analysis
considering biological or technical replicates?
Another fundamental point is that the experimental content appears to be a little too scant for be presented as an original article. It would be preferable to implement the data or change the form of the manuscript (e.g., short communication).
A minor point is that the abstract appears to be too descriptive. Some technical/numerical data should be added.
For these reasons, in my opinion the work needs to be substantially improved and is not ready to be considered for peer-review in its present form.

Reviewer 2 Report
Thank you for the opportunity to review manuscript 2194458. This study presented a biomechanical analysis of custom designed temporomandibular joint (TMJ) prosthesis in combination with a fibular free flap and found that 5-screws configuration has less stress and lower deviation compared with 3-screws configuration.
Generally,the authors did a great and interesting job using Finite Element Analysis to analyze orthopedic biomechanics. Here are some points to consider:
1. The references were not listed in order of appearance in the text.
2. In troduction section
(1) line 54-62. Please concisely summary the result of the listed references.
(2) What is the commonly used ways to reconstruct TMJ? There must have a review in brief.(like rib)
3 M&M section:
(1) line 103-108 /135-142 /149-152 etc. . IS it correct to put this in M&M part? The purpose of the methods section is to describe how you obtained your results. Please reorganize this part carefully and move all the reference review to the discussion part.
(2)line 161-163 These indicators needed to be introduced briefly when they first appeared.
4 When did this patient received the operation? Is there any postoperative functional test performed? (pain? Masticatory efficiency? movement of TMJ? etc.)